

# Valanginian occurrence of Pelomedusoides turtles in northern South America: revision of this hypothesis based on a new fossil remain

Edwin-Alberto Cadena

Facultad de Ciencias Naturales, Grupo de Investigación Paleontología Neotropical Tradicional y Molecular (PaleoNeo), Universidad del Rosario, Bogotá, Colombia
Smithsonian Tropical Research Institute, Panama City, Panama

## ABSTRACT

Pelomedusoides constitutes the most diverse group of Mesozoic and Cenozoic side-necked turtles. However, when it originated is still being poorly known and controversial. Fossil remains from the Early Cretaceous (Valanginian) Rosa Blanca Formation of Colombia were described almost a decade ago as potentially belonging to Podocnemidoidea (a large subclade inside Pelomedusoides) and representing one of the earliest records of this group of turtles. Here, I revise this hypothesis based on a new fragmentary specimen from the Rosa Blanca Formation, represented by a right portion of the shell bridge, including the mesoplastron and most of peripherals 5 to 7. The equidimensional shape of the mesoplatron allows me to support its attribution as belonging to Pelomedusoides, a group to which the previously podocnemidoid material is also attributed here. Although the Valanginian pelomesudoid material from Colombia is still too fragmentary as to be considered the earliest indisputable record of the Pelomedusoides clade, their occurrence is at least in agreement with current molecular phylogenetic hypotheses that suggest they split from Chelidae during the Jurassic and should occur in the Late Jurassic and Early Cretaceous fossil record.

## INTRODUCTION

One of the most diverse clades of Mesozoic and Cenozoic turtles is Pelomedusoides, with fossils worldwide distributed and extant representatives restricted to southern hemisphere (*Ferreira et al., 2018*; *Gaffney et al., 2011*; *Gaffney, Tong & Meylan, 2006*; *Hermanson et al., 2020*; *Vlachos, Randolfe & Sterli, 2018*). Recent molecular phylogenetic hypotheses suggest that they split from Chelidae during the Late Jurassic at 161.7 Ma (149.3–168.9 Ma) (*Pereira et al., 2017*), and total-evidence tip-dating (TE TD) suggest even older date for this splitting during the Early Jurassic at 172.6 Ma (*Holley, Sterli & Basso, 2019*). However, at present, the earliest indisputable fossil record of Pelomedusoides is the bothremydid *Atolchelys lepida* (*Romano et al., 2014*), from the upper Barremian of Brazil; meaning approximately 36 Ma of ghost-lineage.

Corresponding author
Edwin-Alberto Cadena,
edwin.cadena@urosario.edu.co

Almost a decade ago, I described some fragmentary material from the Early Cretaceous (Valanginian) Rosa Blanca Formation of Colombia, which I attributed as potentially belonging to Podocnemidoidea (one of the subclades inside Pelomedusoides) (*Cadena, 2011*). This occurrence has been questioned and considered dubious by *Romano et al. (2014)*, arguing that the presence of an inguinal buttress that medially extends onto the ventral surface of costal 5 is highly variable within Pelomedusoides, even within all of Testudines. Here, I present new material from a locality nearby to the one from where the material described in 2011 came from; from the same segment of the Rosa Blanca Formation (Fig. 1). This new fossil material allows me to revise the hypothesis proposed back in 2011, and present new evidence and comparisons that support the occurrence of Pelomedusoides during the Valanginian in northern South America.

## MATERIALS & METHODS

**Fossil material**. I found the fragmentary material described here in 2016. Recently I added it to the emerging Paleontological Collection of the Facultad de Ciencias Naturales from Universidad del Rosario, in Bogota, Colombia. Its collection identification number is UR-CP-0025. I obtained permit from the Ethics committee of the Universidad del Rosario to execute this study via the DVO005 672-Cv1066 communication. Local landowner Roberto Serrano verbally authorized the paleontological exploration of the zone and collection of the specimen.

**Institutional abbreviations**. CRI, Chelonian Research Institute, Oviedo, Florida, USA; IPN, Museo Geológico Nacional José Royo y Goméz, Bogoté, Colombia; MNHN, Muséum National d'Histoire Naturelle, Paris, France; UR-CP, paleontological collection, Facultad de Ciencias Naturales, Universidad del Rosario, Bogoté, Colombia; USNM, herpetological collection, Smithsonian Natural History Museum, Maryland, USA.

**Carapace length estimation**. In order to establish an estimation of the total length of the carapace to which the fossil fragment belongs, I measured fifteen specimens of extant podocnemidids that I have examined in recent years, as well as the extinct taxa *Francemys gadoufaouaensis* from *Pérez-García (2019)* and *Atolchelys lepida* from *Romano et al. (2014)* (Data S1). Using the software Image J2 (*Rueden, Schindelin & Hiner, 2017*), I set the scale to the one provided in the photos or figures of the specimens and measured the maximum length of both mesoplastra, peripherals 6, 7, and the total length of the carapace (according to their individual preservation). I established the simple linear regression and its equation using Microsoft-Excel (Data S1), and used it to estimate the maximum length of the carapace of UR-CP-0025.

**Comparisons**. For comparisons with other pan-pleurodires including several Pelomedusoides, I created a comparative figure redrawing only the right bridge region from figures or photographs of previous literature or from direct examination of specimens as follow: *Platychelys oberndorferi*, *Notoemys laticentralis*, and *Notoemys zapatocaensis* from *Cadena & Joyce (2015)*; *Notoemys oxfordiensis* from *De La Fuente & Iturralde-Vinent (2001)*; *Francemys gadoufaouaensis* from *Perez-Garcia (2019)*; *Bonapartemys bajobarrealis* from *Lapparent de Broin & de la Fuente (2001)*; *Mendozachelys wichmanni* from *De La*

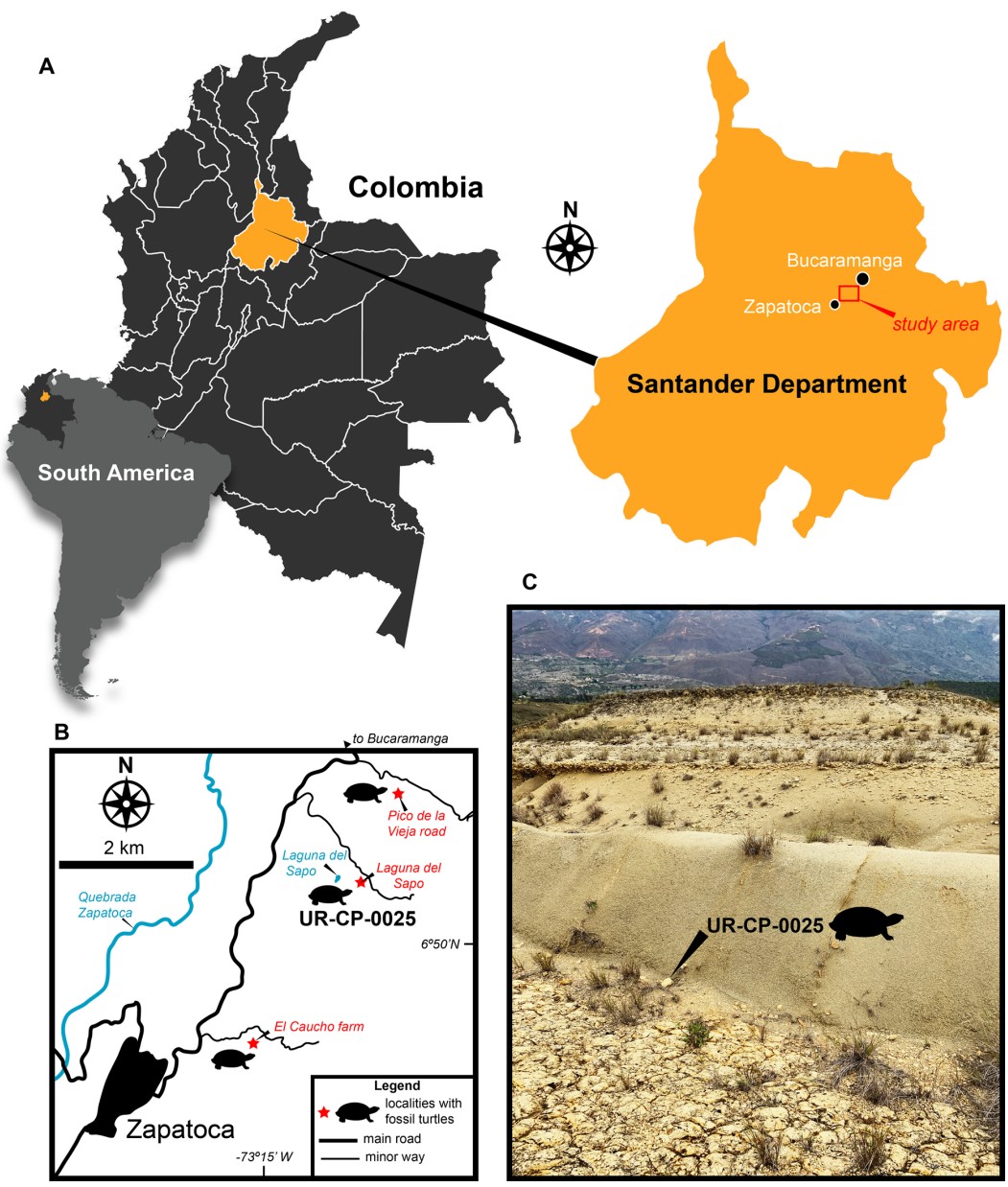

**Figure 1** **Geographic location of UR-CP-0025 Pelomedusoides fossil turtle fragment.** (A) Map of South America, Colombia, and Santander Department, including the study area. (B) Northeast region of Zapatoca showing the three localities from which fossil turtles have been collected: El Caucho Farm, the type locality for *Notoemys zapatocaensis* (*Cadena, Jaramillo & Bloch, 2013*); Pico de la Vieja road, from where IPN 16 EAC-14012003-1A and IPN 16 EAC-14012003-1B (*Cadena, 2011*) referred here to Pelomedusoides came from; and Laguna del Sapo locality from where UR-CP-0025 Pelomedusoides described here came from. (C) Laguna del Sapo locality outcrop showing the discovery of UR-CP-0025 at the base of a calcareous yellow siltstone layer.

*Fuente et al. (2017)*; *Prochelidella cerrobarcinae* from *De La Fuente et al. (2011)*; *Euraxemys essweini* and *Cearachelys placidoi* from *Gaffney, Tong & Meylan (2006)*; *Araripemys barretoi* from *Meylan (1996)*; *Dortoka vasconica* from *Lapparent de Broin & Murelaga (1996)*; *Pelomedusa subrufa* CRI-5200, *Podocnemis expansa* USNM-29476 and *Chelus fimbriata* MNHN-2581A from personal reference photo gallery; and UR-CP-0025 from this study.

## RESULTS

### Systematic paleontology

PLEURODIRA Cope, 1864
PELOMEDUSOIDES Cope, 1868
Incertae Sedis
Fig. 2

Referred material.—UR-CP-0025, a portion of the right shell bridge including the mesoplastron, peripheral 6, portions of peripherals 5 and 7, as well as the lateral most portions of right hyoplastron and hypoplastron. From *Cadena (2011)*: IPN 16 EAC-14012003-1A, left partial costal 5; IPN 16 EAC-14012003-1B, posterior peripheral bone. Locality and Age.—I collected UR-CP-0025 from a locality nearby the Laguna del Sapo (6°50′34″N, −73°14′17.3″W), approximately 1.5 km southwest of the Pico de la Vieja road locality where I found the material reported in *Cadena (2011)* (Figs. 1A–1B). The Laguna del Sapo locality is northeast of Zapatoca, Santander Department, Colombia; and it is part of the recently defined Carrizal Member of the shallow marine Rosa Blanca Formation (*Etayo-Serna & Guzmen-Ospitia, 2019*), correlated to the base of the late Valanginian (∼135 Ma) based on the occurrence of the ammonite *Saynoceras verrucosum*, according to the biochronostratigraphic framework of *Ogg, Ogg & Gradstein (2016)*. I found UR-CP-0025 at the base of a calcareous yellow siltstone layer (Fig. 1C).
Remarks.—UR-CP-0025 is attributed as belonging to Pelomedusoides based on having an equidimensional mesoplastron (Fig. 3). IPN 16 EAC-14012003-1A (*Cadena, 2011*, fig. 1) is also reassigned as potentially belonging to Pelomedusoides based on the massive inguinal buttress scar extending medially onto it, as well as ventrally projected (Character 150, *Gaffney, Tong & Meylan, 2006*). *Romano et al. (2014)* ignored that the character used is dealing with the degree of extension of the inguinal buttress as clearly pointed out in *Cadena (2011* fig. 1) with examples in different Testudines. IPN 16 EAC-14012003-1B is similar in size to the peripherals of UR-CP-0025 described herein, much larger than the peripherals of *Notoemys zapatocaensis* (*Cadena, Jaramillo & Bloch, 2013*), making it also potentially belonging to Pelomesudoides.
**Description.** UR-CP-0025 constitutes a portion of the right shell bridge, preserving the mesoplastron, the most posterolateral corner of the right hyoplastron, the most anterolateral portion of the right hypoplastron, peripheral 6, and portions of peripherals 5 and 7. In ventral view (Figs. 2A–2B), the mesoplastron exhibits an almost equidimensional circular-shape, and it is in lateral and posterolateral contact with peripherals 5 and 6

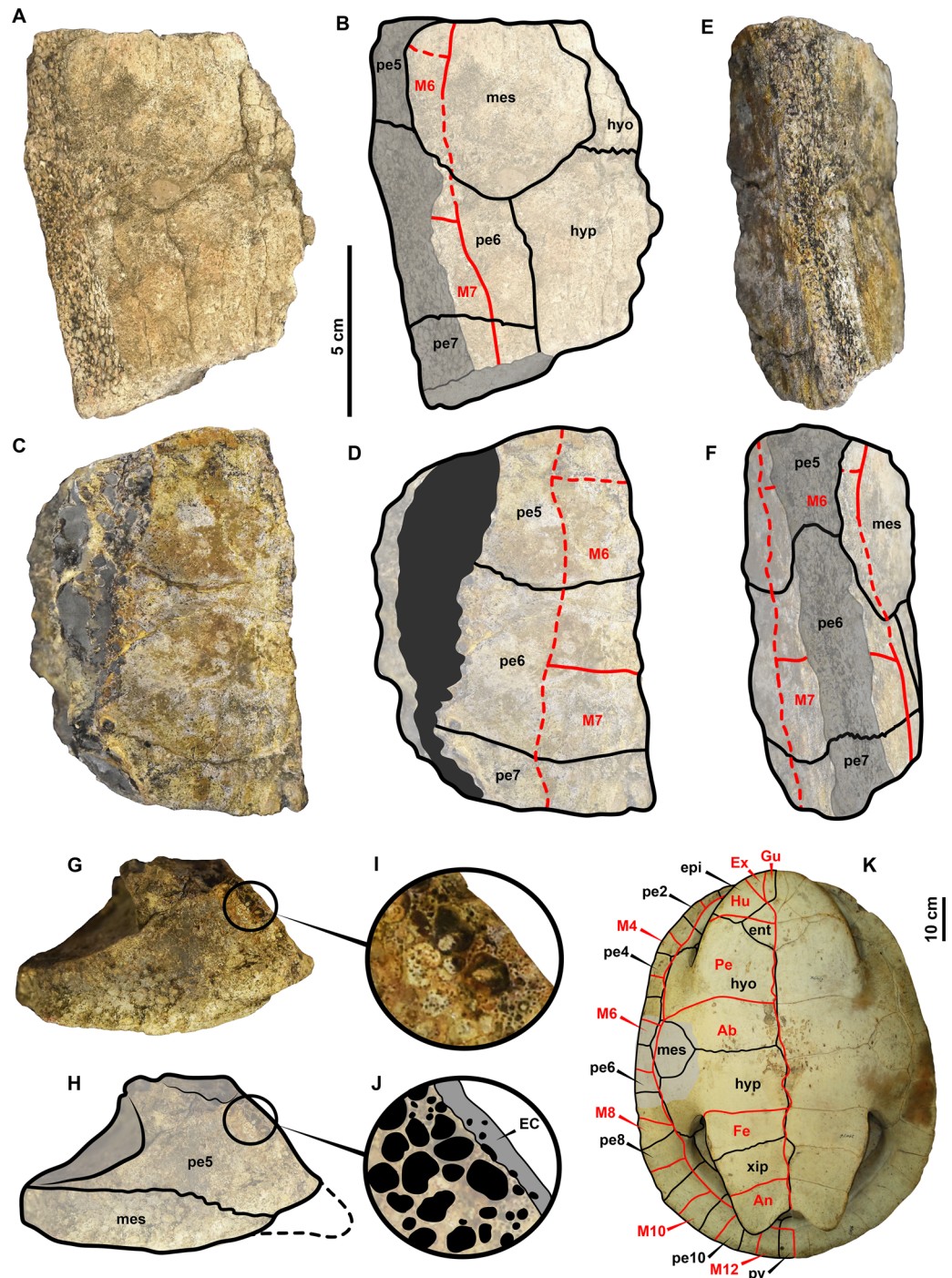

**Figure 2** **UR-CP-0025 Pelomedusoides shell bridge fragment from the Valaginian of Colombia.** (A–B) ventral view. (C–D) dorsal view. (E–F) lateral view. (G–H) anterior view. (I–J) close-up of the margin of peripheral 5, showing the external cortex and cancellous bone. (K) A complete shell in ventral view of *Podocnemis expansa* USNM-29476 specimen, grey region indicates the anatomical corresponding part preserved in UR-CP-0025. Abbreviations: Ab, abdominal scute; An, anal scute; ent, entoplastron; EC, external cortex; epi, epiplastron; Ex, extragular scute; Fe, femoral scute; Gu, gular scute; Hu, humeral scute; hyo, hyoplastron; hyp, hypoplastron; M, marginal scute; mes, mesoplastron; pe, peripheral; py, pygal; xip, xiphiplastron. 10 cm scale bar applies only for (K). Red lines indicate sulci, and dotted lines possible shape and location. Photo credits (A, C, E, D): Andrés Alfonso-Rojas.

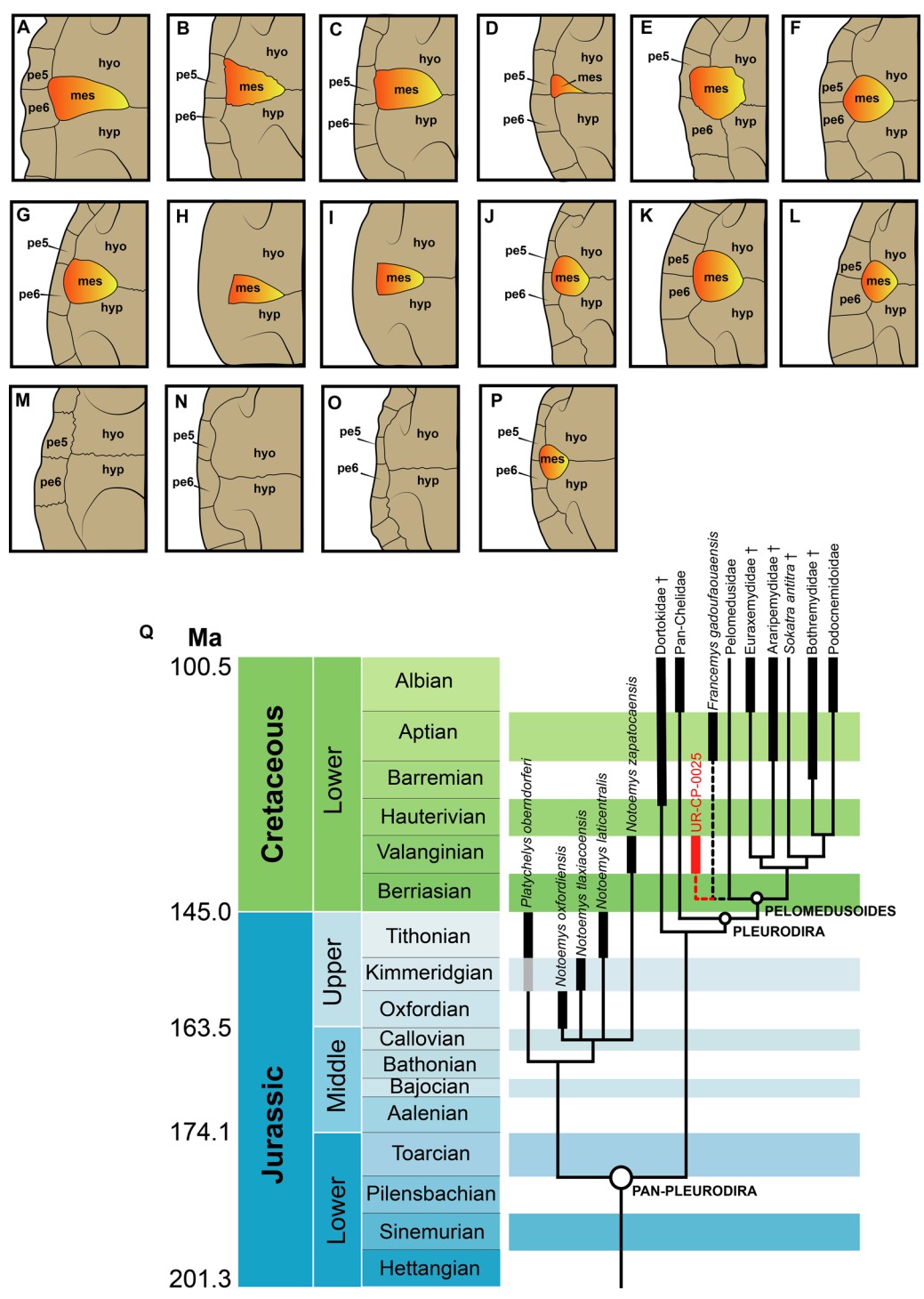

**Figure 3** **Comparisons of the right shell bridge for several pan-pleurodires and their simplified phylogeny.** (A) *Platychelys oberndorferi*. (B) *Notoemys oxfordiensis*. (C) *Notoemys laticentralis*. (D) *Notoemys zapatocaensis*. (E) UR-CP-0025 Pelomedusoides. (F) *Francemys gadoufaouaensis*. (G) *Bonapartemys bajobarrealis*. (H) *Mendozachelys wichmanni*. (I) *Prochelidella cerrobarcinae*. (continued on next page...)

**Figure 3 (…continued)**
(J) *Pelomedusa subrufa* CRI-5200. (K) *Euraxemys essweini.* (L) *Cearachelys placidoi.* (M) *Araripemys barretoi.* (N) *Dortoka vasconica.* (O) *Chelus fimbriata* MNHN-2581A (P) *Podocnemis expansa* USNM-29476. (Q) Simplified phylogeny of Pan-Pleurodira based on *López-Conde et al. (2017)*, *De La Fuente et al. (2017)* and *Hermanson et al. (2020)*, with the potential position of UR-CP-0025 and *Francemys gadoufaouaensis* (*Pérez-García, 2019*); *Atolchelys lepida* (*Romano et al., 2014*) included inside Bothremydidae. Abbreviations: Ma, million years; mes, mesoplastron; hyo, hyoplastron; hyp, hypoplastron; pe, peripheral. See methods for full references of the taxa illustrated here. Right mesoplastron highlighted in orange-yellow.

respectively. Medially and posteromedially is in contact with the right hyoplastron and right hypoplastron respectively. The most lateral portion of peripherals 5 to 7 is missing (natural breaking). Also in this view, there is evidence of some of the sulci, particularly between marginals and of these with the abdominal scute. There is not indication that the pectoroabdominal sulcus reached the anterolateral corner of mesoplastron. In dorsal and lateral views (Figs. 2C–2F), the peripheral 6 is the most complete of the three preserved, showing a rectangular shape with its most medial margin (contact with costals) missing. The sulci between marginals are poorly preserved, however there is enough evidence that they were restricted to the peripherals, without reaching the costoperipheral sutural margin. The sutural contact between peripherals shows a medial indentation (Figs. 2E–2F), however this seems to be due to that the bone is naturally cut and cancellous tissue exposed. In anterior view (Figs. 2G–2H), the peripheral 5 and mesoplastron contact is well defined, and the bridge angle formed between the peripherals and the plastron indicates that the shell was probably low to moderate dome-shaped. Also in this view is evident the considerable thickness of these bones. A close-up of the margin of peripheral 5 (Figs. 2I–2J) shows a very thin external bone cortex and abundance of large pores at the cancellous bone. A large (88.17 cm, carapace length) of the extant *Podocnemis expansa* USNM-29476 is shown in Fig. 2K for comparison and anatomical location of UR-CP-0025 in a turtle shell.

## DISCUSSION

**Mesoplastra of Pan-Pleurodira**. The mesoplastra bones have exhibited important modifications along turtle evolution. In basal Pan-Testudines as for example *Odontochelys semitestacea* they were two separate bony plates meeting medially (*Li et al., 2008*). In basal Testudines as for example *Kayentachelys áprix* there was a reduction in the number of mesoplastra, being only one pair extended medially reaching the central fontanelle (*Joyce, 2007*, fig 11). Another transformation of mesoplastra occurred in the both major groups of turtles, with their complete lost in Cryptodires and being one pair but they do not contact one another medially in Pan-Pleurodira (*Cadena & Joyce, 2015*; *Joyce, 2007*). Inside Pan-Pleurodira the mesoplastra have exhibited additional transformations from the primitive condition exhibited by Platychelyidae and Cretaceous-Paleocene members of Pan-Chelidae (Figs. 3A–3C, 3G–3I) of being almost triangular in shape, much wider than long to the condition exhibited by almost all Pelomedusoides of having almost equidimensional mesoplastra (Figs. 3E–3F, 3J–3L, 3P). An equidimensional mesoplastron was considered by *Gaffney, Tong & Meylan (2006)* as characteristic of a Nanorder that they defined as Eupleurodira (Cheloides = Pan-Chelidae plus Pelomedusoides). As

I show in Fig. 3, the condition in pan-chelids who have mesoplastra is similar to the one exhibited by platychelids, which allow me to suggest that instead this is a characteristic of Pelomedusoides, shared by UR-CP-0025 described herein (Fig. 3E). Another transformation of mesoplastra inside Pan-Pleurodira is their complete lost in some Dortokidae, some Araripemydidae, crown-Chelidae, and *Pelusios* spp. (*Gaffney, Tong & Meylan, 2006*) (Figs. 3M–3O).

**Carapace size estimation of UR-CP-0025.** Using the simple linear regression equation ($y = 6.5362x + 5.5002$, $x$ corresponding to maximum mesoplastron length, and $y$ maximum carapace length) obtained from specimens of extant podocnemidids and some fossil pelomedusoids (Data S1). I estimated that the length of the carapace of UR-CP-0025 was of ∼40.59 cm, indicating a much larger size in contrast to the exhibited by Jurassic and Early Cretaceous platychelids, which fluctuated between 20 to 27 cm (*Cadena, Jaramillo & Bloch, 2013*, table 8.1). This suggests that the increase in shell size was a characteristic exhibited by early representatives of Pelomedusoides; a trend that continued during the Late Cretaceous (*Hermanson, Ferreira & Langer, 2016*) and the Cenozoic, with the giant pelomedusoids from the Paleocene of Colombia (*Cadena, Bloch & Jaramillo, 2012*; *Cadena et al., 2012*), and the Miocene *Stupendemys geographicus* from northern South America (*Cadena et al., 2020*).

**Implications of UR-CP-0025 for Pelomedusoides history understanding.** With the description of UR-CP-0025 and its attribution as belonging to Pelomedusoides (see above), I show that they inhabited northern South America during the Early Cretaceous. A hypothesis that is in agreement with recent molecular phylogenetic hypotheses that suggest they split from Chelidae during the Jurassic (*Holley, Sterli & Basso, 2019*; *Pereira et al., 2017*), therefore their fossil record should be expected to occur in Late Jurassic and Early Cretaceous sequences (Fig. 3Q). However, it is important to point out that UR-CP-0025 and the material previously described also from Rosa Blanca Formation (*Cadena, 2011*) are still too fragmentary to be recognized as the earliest indisputable record of the group, which it is not intention of this study. With this study, I once again showed that the Rosa Blanca Formation is still being a very productive and promising rock sequence in northern South America for future paleontological studies and the understanding of the Early Cretaceous faunas, including the evolution of Pelomedusoides turtles.

## CONCLUSIONS

With this study, I once again showed that the Rosa Blanca Formation is still being a very productive and promising rock sequence in northern South America for future paleontological studies and the understanding of the Early Cretaceous faunas, including the evolution of Pelomedusoides turtles. Despite the taxonomic uncertainty that fragmentary material as UR-CP-0025 has, it exhibits the characteristic equidimensional mesoplastron of Pelomedusoides turtles, supporting the occurrence of this group of turtles in northern South America during the Early Cretaceous.

## ACKNOWLEDGEMENTS

Thanks to A Alfonso-Rojas for help in cataloging and photographing the specimen. Thanks to P Pritchard (R.I.P) (Chelonian Research Institute), F. Lapparent de Broin (Muséum National d'Histoire Naturelle), O Castaío (Instituto de Ciencias Naturales) and staff at the Smithsonian Natural History Museum for access to the collections. Thanks to R Serrano for access to the zone where the specimen was collected. Thanks to the three reviewers; M, De La Fuente, A Perez Garcia, and N, Benevenuto for the insightful comments, as well as *PeerJ* editor M Young.

### Funding

This study was supported by Universidad del Rosario, Fondos de Arranque 2018 (code IV-TFA022). The funders had no role in study design, data collection and analysis, decision to publish, or preparation of the manuscript.

### Grant Disclosures

The following grant information was disclosed by the author:
Universidad del Rosario, Fondos de Arranque 2018 (code IV-TFA022).

### Competing Interests

The author declares that he has no competing interests.

### Author Contributions

- Edwin-Alberto Cadena conceived and designed the experiments, performed the experiments, analyzed the data, prepared figures and/or tables, authored or reviewed drafts of the paper, and approved the final draft.

### Animal Ethics

The following information was supplied relating to ethical approvals (i.e., approving body and any reference numbers):

The Ethics Committee of Universidad del Rosario gave aval for the development of this project under document number DVO005 672-Cv1066.

### Field Study Permissions

The following information was supplied relating to field study approvals (i.e., approving body and any reference numbers):

Local landowner Roberto Serrano verbally authorized the paleontological exploration of the zone and collection of the specimen.

### Data Availability

The raw measurements are available as a Supplemental File.

## Supplemental Information

Supplemental information for this article can be found online at http://dx.doi.org/10.7717/peerj.9810#supplemental-information.

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
