# Peer review of "Valanginian occurrence of Pelomedusoides turtles in northern South America: revision of this hypothesis based on a new fossil remain"

_PeerJ, doi:10.7717/peerj.9810_

## Round 0.1 · original submission · Minor Revisions

Dear authors,

Based on the reviewers comments I have made the decision of ‘minor revisions’.

Please note two reviewers have commented on the linear regression analysis, and the small sample size it is based on. I agree, and that is one change that will need to be made prior to re-submission.

I look forward to receiving your revised manuscript.

·

Basic reporting

The manuscript entitled “Valanginian occurrence of Pelomedusoids turtles in northern South America: Revision of this hypothesis based on a new fossils remains” is concerned with the description of a new shell fragment of turtle from Rosablanca Formation (Early Cretaceous, Colombia). This discovery allows the author assigns to the described specimen to Pelomedusoides. Overall, this brief article is a good contribution with nice figures and interesting data. The text is clear and well written and the references are updated.

Experimental design

Previously, on the basis of fragmentary carapace bones (one left costal 5 and a left peripheral 9?) recovered from Rosablanca Formation (Late Valanginian) at northeast of Zapatoca town, Colombia, Cadena (2011) suggested the assignation of these carapace remains to Podocnemidoidea taking into account the inguinal buttress that medially extends onto the visceral surface of the costal bone 5. Lately, Romano et al (2014) suggested that this character is highly variable between Pelomedusoids and in Testudines. Considering these carapace fragments from Early Cretaceous Colombia as undiagnostic

The new discovery of a shell fragment (a right portion of the shell bridge including a mesoplastron and peripherals 5-7) from Rosablanca Formation allows to Edwin Cadena to attribute this specimen to Pelomedusoides Incertae Sedis on the basis of proportions and shape of mesoplastron bone. I concur with this determination because previously Lapparrent de Broin (2000) in her phylogenetic analysis of Pelomedusoids recognized a rounded and lateral mesoplastron as a derived character of this pleurodiran clade.

The fossil material (UR-CP-0025) studied by Cadena is properly housed in Paleontological Collection of the Facultad de Ciencias Naturales from Universidad del Rosario, in Bogotá. The author for carapace estimation length of the fragmentary exemplar of Rosablanca Formation (UR-CP-0025) used two large Podocnemis expansa specimens and he set the scale to the one provided in the photos of the specimens
and measured the maximum length of both mesoplastra, peripherals 6, 7, and the total length of the carapace, and he established a simple linear regression. For comparative
purposes, the author illustrated the right portion of the shell of UR-CP-0025 and several extinct and extant panpleurodiran species.

Validity of the findings

Although the described turtle material is so fragmentary, preventing a detailed taxonomic determination, it is relevant due to the Valanginian mesoplastron outline and proportions suggest Pelomedusoids affinities, this record represents the earliest Pelomedusoids worldwide.

Additional comments

The manuscript entitled “Valanginian occurrence of Pelomedusoids turtles in northern South America: Revision of this hypothesis based on a new fossils remains” is concerned with the description of a new shell fragment of turtle from Rosablanca Formation (Early Cretaceous, Colombia). This discovery allows the author assigns to the described specimen to Pelomedusoides. Overall, this brief article is a good contribution with nice figures and interesting data. The text is clear and well written and the references are updated.
Previously, on the basis of fragmentary carapace bones (one left costal 5 and a left peripheral 9?) recovered from Rosablanca Formation (Late Valanginian) at the northeast of Zapatoca town, Colombia, Cadena (2011) suggested the assignation of these carapace remains to Podocnemidoidea taking into account the inguinal buttress that medially extends onto the visceral surface of the costal bone 5. Lately, Romano et al (2014) suggested that this character is highly variable between Pelomedusoids and in Testudines. Considering these carapace fragments from Early Cretaceous Colombia as undiagnostic

The new discovery of a shell fragment (a right portion of the shell bridge including a mesoplastron and peripherals 5-7) from Rosablanca Formation allows to Edwin Cadena to attribute this specimen to Pelomedusoides Incertae Sedis on the basis of proportions and shape of mesoplastron bone. I concur with this determination because previously Lapparrent de Broin (2000) in her phylogenetic analysis of Pelomedusoids recognized a rounded and lateral mesoplastron as a derived character of this pleurodiran clade.

The fossil material (UR-CP-0025) studied by Cadena is properly housed in Paleontological Collection of the Facultad de Ciencias Naturales from Universidad del Rosario, in Bogotá. The author for carapace estimation length of the fragmentary exemplar of Rosablanca Formation (UR-CP-0025) used two large Podocnemis expansa specimens and he set the scale to the one provided in the photos of the specimens and measured the maximum length of both mesoplastra, peripherals 6, 7, and the total length of the carapace, and he established a simple linear regression. For comparative
purposes, the author illustrated the right portion of the shell of UR-CP-0025 and several extinct and extant panpleurodiran species.


Although the described turtle material is so fragmentary, preventing a detailed taxonomic determination, it is relevant due to the Valanginian mesoplastron outline and proportions suggest Pelomedusoids affinities, this record represents the earliest Pelomedusoids worldwide.

Minor comments
In lines 149-150 Edwin Cadenan says “… the primitive condition exhibited by Platychelyidae and Cretaceous members of Pan-Chelidae (Fig. 3A–C, G–I)” should said “ … the primitive condition exhibited by Platychelyidae, Cretaceous and Paleocene members of Pan-Chelidae (Fig. 3A–C, G–I)” (see Paleocene panchelids with mesoplastra in Bona, 2006, Bona and de la Fuente, 2005)


In line 217, in reference list, Edwin Cadena says “ de la Fuente, MS, Umazana, AM , Sterli J, and Carballido, J.L. 2011…”, should said “ de la Fuente, MS, Umazano, AM , Sterli J, and Carballido, J.L. 2011…”

·

Basic reporting

This is an interesting and well-written manuscript. Several minor suggestions are indicated here (see general comments)

Experimental design

No comment

Validity of the findings

No comment

Additional comments

This is an interesting and well-written manuscript. Several minor suggestions are indicated here:

- I suggest considering the geographical figure as Figure 1 instead of 2, and citing it for the first time in the last paragraph of the Introduction.
- I do not understand why the carapace length estimation is limited to the comparison with a single extant form. The shell morphology of the Valanginian taxon is unknown. Therefore, I suggest also add comparisons with other pleurodires with different proportions and morphologies of the carapace. In this sense, I would also include extinct forms (as is the case, among others, of Francemys, recognized in this work as having a similar evolutionary degree).
- I suggest moving the statement indicated as “Remarks” in the “Systematic Paleontology” section to the “Dicussion”, and justifying it with references.
- The author indicates “Another transformation of mesoplastra inside Pan-Pleurodira is their complete lost in Dortokidae, Araripemydidae”. However, I suggest changing this for “… complete lost in some Dortokidae, some Araripemydidae”: mesoplastra are present in the dortokid Eodortoka and in the araripemydid Taquetochelys.
- Figure 1: I suggest using a dashed line to represent the broken edges: in the current version of the drawings it is not possible to know if, for example, the anterior margin of the mesoplastron, or the lateral margin of the peripherals, are or are not broken.
- Caption of Fig. 1: Please, complete the following sentence, indicating the age and locality: “UR-CP-0025 Pelomedusoides shell bridge fragment FROM…”
- Caption of Fig. 2: Please also add more information in the sentence “Geographic location of the fossil finding STUDIES HERE, CORRESPONDING TO… FROM…”
- Figure 2: I suggest a separation between the map of South America and that of Colombia like that between those of Colombia and the Santander Department.

I’m available for any additional clarification that author and editors may need.

Sincerely,

Adán

--
Dr. Adán Pérez-García

http://dfmf.uned.es/biologia/personal/aperez/

Grupo de Biología Evolutiva, Facultad de Ciencias, UNED
Madrid, Spain

·

Basic reporting

No comment.

Experimental design

No comment.

Validity of the findings

The data that have been provided are not robust: a simple linear regression using only two specimens do not provide any statistical power. There is a huge confidence interval, resulting in a unreliable estimate. The linear model for the estimate is imprudent. To improve this, I suggest remove shell size stimatives from manuscript or redo the analysis by measuring as many especimens as possible.

Additional comments

Comments and suggestions throughout the manuscript are attached in the following PDF document: peerj-reviewing-49823v1.

---

## Round 0.2 · accepted · Accept

Dear authors,

I am happy to accept your manuscript for publication.

The production staff will be in contact about the proof stage of your manuscript.

Thank you again for choosing PeerJ as your publication venue, and I hope you will use us again in the future.

·

Basic reporting

This manuscript contains relevant information about the early history of Pelomedusoides. In addition, it is well structured, with a well-written text and good images that provide clear comparative osteology.

Experimental design

No comment.

Validity of the findings

The description of the material is consistent with that illustrated and compared in the manuscript. Furthermore, the author improved the data by measuring new specimens, as suggested. The conclusions are clear and have now been pointed out after the discussion.

Additional comments

This manuscript contains relevant information about the early history of Pelomedusoides. In addition, it is well structured, with a well-written text and good images that provide clear comparative osteology.

The description of the material is consistent with that illustrated and compared in the manuscript. Furthermore, the author improved the data by measuring new specimens, as suggested. The conclusions are clear and have now been pointed out after the discussion.

I congratulate the author for the excellent work and I am available for any clarification.

Sincerely,

Natália.